# Multimorbidity and depression among older adults in India: Mediating role of functional and behavioural health

Salmaan Ansari[1], Abhishek Anand[2]*, Babul Hossain[3]

1 Department of Population Policies & Programs, International Institute for Population, Sciences, Mumbai, India, 2 Department of Public Health and Mortality Studies, International Institute for Population, Sciences, Mumbai, India, 3 Department of Development Studies, International Institute for Population Sciences, Mumbai, India

* anandabhishek361@gmail.com

**Data Availability Statement:** The data is available on request from website of International Institute for Population Sciences, Mumbai. https://www.iipsindia.ac.in/content/LASI-data.

**Funding:** The author(s) received no specific funding for this work.

## Abstract

Researchers have long been concerned about the association between depression and the prevalence of multiple chronic diseases or multimorbidity in older persons. However, the underlying pathway or mechanism in the multimorbidity-depression relationship is still unknown. Data were extracted from a baseline survey of the Longitudinal Ageing Survey of India (LASI) conducted during 2017–18 (N = 31,464; aged $\geq$ 60 years). Depression was assessed using the 10-item Centre for Epidemiological Studies Depression Scale (CES-D-10). Multivariable logistic regression was used to examine the association. The Karlson–Holm–Breen (KHB) method was adopted for mediation analysis. The prevalence of depression among older adults was nearly 29% (men: 26% and women 31%). Unadjusted and adjusted estimates in binary logistic regression models suggested an association between multimorbidity and depression (UOR = 1.28; 95% CIs 1.27–1.44 and AOR = 1.12; 95% CIs 1.12–1.45). The association was particularly slightly strong in the older men. In addition, the association was mediated by functional health such as Self Rated Health (SRH) (proportion mediated: 40%), poor sleep (35.15%), IADL disability (22.65%), ADL disability (21.49%), pain (7.92%) and by behavioral health such as physical inactivity (2.28%). However, the mediating proportion was higher among older women as compared to older men. Physical inactivity was not found to be significant mediator for older women. The findings of this population-based study revealed that older people with multimorbidity are more likely to suffer depressive symptoms in older ages, suggesting the need for more chronic disease management and research. Multimorbidity and depression may be mediated by certain functional health factors, especially in older women. Further longitudinal research is needed to better understand the underlying mechanisms of this association so that future preventive initiatives may be properly guided.

**Competing interests:** The authors have declared that no competing interests exist.

## Introduction

Age-related multimorbidity is an escalating issue that poses a major challenge for health care systems worldwide [1]. Multimorbidity can be explained as the presence of two or more chronic physical conditions and its prevalence has increased in the last decades mainly because of ageing and increased longevity [2–4]. Multimorbidity is linked with a high prevalence of mortality and it also has a severe impact on the social and emotional wellbeing of the individuals [5, 6]. The prevalence of multimorbidity varies significantly in low-middle income countries, it's prevalence in South Asian countries such as India and Bangladesh, ranges from 4.5 percent to 83 percent [7]. The figures are also high in developed nations. Few studies have shown high national level estimates of multimorbidity in Spain and Germany, approximately 60 percent for individuals age 65 years and above [8, 9]. Population prevalence studies in Australia, Canada, Spain and several European countries have also reported high prevalence of multimorbidity [4, 10, 11]. Meanwhile, depression in older ages is a public health complication that cannot be ignored [12]. It is also strongly associated with various morbidity conditions, increased risk of self-harm, declining social and cognitive functioning. Based on large-scale surveys, the prevalence of depression is estimated between 10 to 15 percent, and its contribution to the global burden of disease has surged in the last few decades [12, 13].

While, India, the second most populated country, has recently seen a rapid demographic change, with a rising number of older individuals in the overall population, multimorbidity and depression pose serious public health challenges in India. The recent estimation suggest that almost one out of four Indian individual reported having multimorbidity while 3.3 percent had depressive disorders [7, 12, 14]. While looking at Indian scenario, the burden of depression has been soaring issue among adults and particularly among later ages. Depressive disorders affect around 46 million people in India and it contributes majorly to DALYs (disability-adjusted life-years) [15]. According to the Global Burden of Disease Study, people with multimorbidity are at higher risk for developing depressive symptoms compared to those without any morbidity [16, 17].

The increasing occurrence of chronic conditions requires an integrative approach in assessing the impact of multimorbidity on the older population. Several literatures have documented that depression is usually less prevalent in the older population in comparison with people of younger ages, however, they suffer severe consequences [18, 19]. This leads to poor quality of life among the older population as well as increased mortality risk [19]. It is evident from earlier studies that individuals with major chronic physical conditions had higher depression level and the prevalence of depression was mostly found among people having hypertension [20]. Patients with cardiovascular diseases have a higher risk of depression when compared with the general population [21], similarly, diabetes, kidney disease, cancer, or lung disease are major risk factors for depression [22–24].

In addition, functional health has an integral outcome of the ageing process, which is directly related to the quality of life or wellbeing in older ages. It is also important to note that poor functional health can further degrade the mental health status of older adults. It is not only limited to physical activities, but a combination of physical, social, and cognitive abilities. "World Health Organization" (WHO) on ageing documented that loss of these functions has a severe impact on the quality of life and health status of older adults [1]. Further, evidence suggested that functional health may influences the relationship between multimorbidity and depression in older adults. The functional health might act as a mediator between an association of multimorbidity and depression by limiting the social and physical abilities of individuals [25, 26], as multimorbidity escalates the deterioration of physical health leading to adverse emotional outcomes which facilitate depressive symptoms [27, 28]. In addition to functional

health risk factors, behavioural health issues such as alcohol consumption, smoking, physical activity are also linked with high depressive disorders [29–31]. Alcohol intake has harmful effects on many diseases and several most prevalent multimorbid conditions such as hypertension, diabetes and heart diseases are exacerbated by alcohol consumption [32]. Physical inactivity is another unhealthy behaviour which is associated with depressive symptoms [33]. High physical activity is more effective in reducing depression in comparison with low activity or inactivity [34]. These behavioural health issues may predispose to the development of depression and then to multimorbidity.

While gender becomes crucial factor determining health of an individual and healthy ageing. Evidence suggests that women have a higher life expectancy, and experiences poor health in older ages [1, 35]. As a result, older women are more vulnerable to morbidity than men because of the higher prevalence of acute, chronic, and mental disorders [28, 29, 36]. Studies have shown that women have a larger disease burden and the prevalence of chronic conditions increase with their age [37]. According to a multi-country study, sociocultural inequities between men and women, in addition to biological characteristics, are determining factors in health of older people [38]. It is apparent from studies that women live more years with morbidities mainly in a late stage of their life. Clustering of diseases creates problem in designing adequate interventions and preventive strategies, and due to gender differences, there is a need to study existing underlying multimorbidity pattern of a population accordingly. Thus, analysing sex differences in the association between multimorbidity and depression will highlight the key factors explaining the disparities between older men and women.

Meanwhile, India with 1.2 billion people in the world in 2011, and people of age 60 years and above accounted for 8.6 percent of the total population shows a significant greying process of demography for the country. It is estimated that there will be about 319 million older population in India by 2050 [39]. In the Indian context, the majority of studies on health problems and ageing have focussed on the consequences of individual diseases among older adults. For instance, studies by Chauhan et al. (2021) and Srivastava et al. (2021) have evaluated the repercussions of living arrangements and other behaviour health factors on depression among older adults in India [40, 41]. A community level study on primary care population in India estimated high prevalence of multimorbidity among older adults, women and individuals of high socioeconomic status, and significant gender differences in prevalence of multimorbidity have been reported [42, 43]. While a study by Hossain et al. (2021) have explored the association between physical limitations and self-reported depressive symptoms among Indian older adults considering marital status as a moderator in such association [44]. However, despite the fact that there is an increased chance of depression in older persons as the number of chronic illnesses or multimorbidity grows, the underlying route or mechanism explaining the multimorbidity-depression association remains unexplained in the Indian context.

Our study fills a gap in the existing literature by identifying potential mediators in the link between multimorbidity and depression among older adults. In addition, this study also looks into gender differences in the mediation mechanism of the link between multimorbidity and depression. We state the following hypotheses: first, multimorbidity is positively associated with depression in older men and women in India. Second, factors selected to functional and behavioural health would exert an indirect effect on the relation between multimorbidity and depressive symptoms in older men and women in India.

Fig 1 shows the preliminary theoretical model of the association between multimorbidity, and depression shows that multimorbidity might directly influence depression, but might also indirectly influence depression through the selected potential mediators.

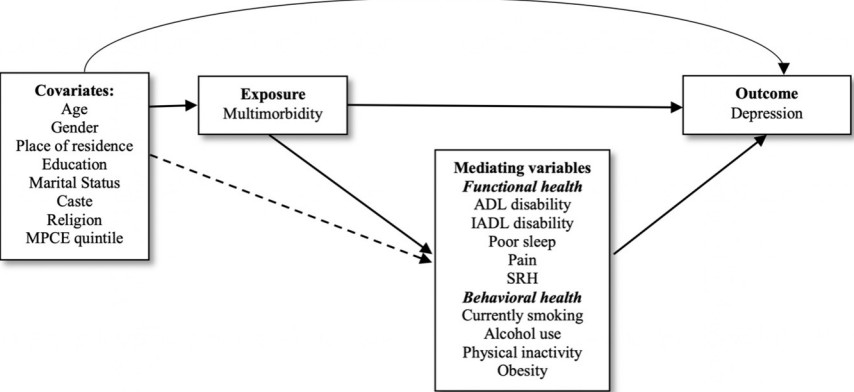

**Fig 1. Preliminary theoretical model of the multimorbidity-depression association and its potential mediators.**

## Methods

### Data source

This study was based on a cross-sectional design of the data from the first wave of the Longitudinal Ageing Study in India (LASI) conducted during 2017–18, an ongoing nationally representative longitudinal study involving people aged 45 and over in India. Its objective is to determine the health, economic, psychological, and social wellbeing of older people aged 45 or above in India. A baseline survey of this large-scale survey was conducted from April 2017 to December 2018, and an individual will be followed every 2 years for the next 25 years. At the time the current study was undertaken, no follow-up data were available. The baseline survey including 72,250 older adults aged 45 and above and their partners (regardless of age) represents all the states (except Sikkim) and Union Territories (UTs) across the country. All participants provided informed consent to ensure anonymity and to tell them about the survey's goals. The LASI adopted a multistage stratified area probability cluster sampling design for selecting the sample across the country. The current study consisted of 31,464 older people aged 60 years and above.

The necessary guidance and approval for data collection in LASI survey was approved by Indian Council of Medical Research (ICMR). The written informed consent was taken from each household and age-eligible individual. In accordance with human subject protection, four consent forms were used, household informed consent, individual informed consent, consent for blood samples collection for storage and future use (DBS), and proxy consent.

**Outcome variable: Depressive symptoms.** In this study, the main outcome of interest was depressive symptoms. It was assessed using the Centre for Epidemiologic Studies Depression Scale (CES-D-10) with four scale option categories from (1) rarely or never (< 1 day) to (4) most or all of the time (5–7 days). Ten different questions were asked about the experience of participants during the past week: trouble concentrating, feeling depressed, low energy, fear of something, feeling alone, bothered by things, everything is an effort, feeling happy, hopeful, and satisfied. On this 10-items scale first seven were based on negative symptoms, while the remaining three were based on positive symptoms. Therefore, a zero score was given to those who have responded to negative symptoms as "rarely or never (< 1 day)", and "sometimes (1 or 2 days)", while the remaining two categories were coded as one. However, the scoring was reversed in the case of positive symptoms. The composite score ranged from 0 to 10, and those

with a score of four or more were considered to have depression symptoms [45, 46]. Cronbach's alpha indicated that CES-D-10 has excellent reliability (0.8).

**Exposure variable: Multimorbidity.** LASI survey covered 9 different chronic illness: (1) hypertension, (2) chronic heart diseases, (3) stroke, (4) any chronic lung disease, (5) diabetes, (6) cancer or malignant tumour, (7) any bone/joint disease, (8) any neurological/psychiatric disease, and (9) high cholesterol. All chronic illnesses were assessed by asking the following question: "Has any health professional ever diagnosed you with the following chronic conditions or diseases?". Multimorbidity is defined as an individual who has been diagnosed with two or more chronic illnesses [3, 47]. If an individual has one chronic illness or none at all, they are coded as 0 "No multimorbidity," and if they have two or more chronic illnesses, they are coded as 1 "Multimorbidity" (Cronbach Alpha:0.74) [48].

**Mediating variables.** *Activities of Daily Livings (ADL) and Instrumental Activities of Daily Livings (IADL) disability*. ADL and IADL disability were assessed by a series of six and seven different questions respectively. For ADL, participants were asked if they had experienced limitations in any of the following everyday activities: difficulty with dressing, walking across the room, bathing, eating, getting in or out of bed, and using the toilet (including getting up and down). For IADL, participants were asked if they had any difficulty doing any of the following activities: preparing a hot meal, shopping for groceries, making a telephone call, taking medications, doing work around the house or garden, managing money (such as paying bills and keeping track of expenses), and getting around or finding an address in an unfamiliar place. All of the difficulties were addressed, which had been the limitation for over three months. Two separate dichotomous variables were created, with a code of "1" given if the participants reported any of the limitations and "0" if the participants reported no limitations (Cronbach Alpha:0.85) [48–50].

*Poor sleep*. Poor sleep was assessed using three questions being used in prior studies: "How often do you have trouble falling asleep?"; "How often did you wake up during the night and had trouble getting back to sleep?"; and "How often did you wake up too early in the morning and were not being able to fall asleep again?" [51]. Individual questions are grouped into four different responses to measure the frequency of each of these, such as never, rarely (1–2 nights per week), occasionally (3–4 nights per week), and frequently (5 or more nights per week). Poor sleep was coded as 1 "Yes" if participants responded "sometimes (3–4 nights/week)" or "often (5 or more nights/week)" and if they said "Never" or "Rarely (1–2 nights/week)" was coded as 0 "No".

*Pain*. Pain was defined by asking a question to the participants whether they are often troubled with pain. Pain was coded as 1 if they said "Yes," and as 0 if they said "No" [52].

*Self-Rated Health (SRH)*. SRH was defined by using a single question "Overall, how is your health in general?" with five responses categories such as 1 "Very good", 2 "Good", 3 "Fair", 4 "Poor", and 5 "Very poor". Those who responded "Very good", "Good", or "Fair" were coded as 0 "Good" health whereas those who responded "Very poor" or "Poor" were coded as 1 "Poor" Health [53, 54].

*Current smoking*. Smoking status was measured by asking the following two questions: "Have you ever smoked tobacco (cigarette, bidi, cigar, hookah, cheroot) or used smokeless tobacco (such as chewing tobacco, gutka, pan masala, etc.)"; and "Do you currently smoke any tobacco products (cigarettes, bidis, cigars, hookah, cheroot, etc.)". Participants who reported smoking currently were classified as "1" yes, whereas those who had never smoked, or discontinued smoking were classified as "0" no.

*Alcohol use*. Alcohol use was measured based on whether participants consumed alcohol in the previous three months. A dummy-coded variable was created with 0 (representing never

consumed alcohol or any alcohol use in the previous three months) and 1 (representing alcohol consumption in the previous three months).

*Physical inactivity.* Physical activity was assessed by asking the question "How often do you take part in sports or activities that are moderately energetic such as, cleaning house, washing clothes by hand, fetching water or wood, drawing water from a well, gardening, bicycling at a regular pace, walking at a moderate pace, dancing, floor or stretching exercises?". A participant was considered to be physically active if they responded "every day", "more than once a week", "once a week", or "one to three times in a month", whereas those who had responded "never" was considered to be physically inactive. A dichotomous variable was created to define physical inactivity, with 1 representing *physically inactive* and 0 representing *physically active* [55].

*Obesity.* Body Mass Index (BMI) was calculated based on their self-reported weight and height data, as weight in kilograms divided by height in meters squared. The current study employed a standard cut-off for obesity (BMI $\geq$ 30) based on WHO classification [56]. A dichotomous variable was created to measure obesity status in older people and coded as 1 "obese" and 0 "Not obese".

*Covariates.* Different socio-demographic variables were used as covariates in the present study. These are age (60–69 years old, 70–79 years old, $\geq$ 80 years old), educational attainment (no education, less than 5 years, 5–9 years, 10 or more years), marital status (currently in marital union, not in a marital union), caste (Scheduled Caste/Scheduled Tribe, Other Backward Class, and Others), religion (Hindu, Muslim, Others), place of residence (Rural, Urban), and monthly capita per expenditure (MPCE) quintile (Poor, Middle, Rich).

## Statistical analysis

Descriptive statistics and bivariate estimations were performed to describe the study sample. Multivariable binary logistic regressions were used to evaluate the association between multimorbidity and depression among older people. We, therefore, performed regressions stratified by gender to further investigates the gender differentials in the association. Three different separate models for the overall sample, older men and women were run to quantify the association between multimorbidity and depression: an unadjusted model, an adjusted model which only controlled the socio-demographic variables, and an adjusted model which controlled functional and health behaviour characteristics, as well as the socio-demographic characteristics. Results of the regression models are presented as the odds ratios (ORs) with 95% confidence intervals (CIs). To test multicollinearity, the variance inflation factor (VIF) was computed using multiple linear regression taking dependent variable (i.e., depressive symptoms) on a numeric scale [57]. The largest variance found was 2.70, indicating that there was no concern with multicollinearity, especially for the mediators.

Mediation analysis was performed to evaluate the underlying mechanisms of different mediators that may explain the association between multimorbidity (exposure) and depression (outcome) among older people. We performed a mediation analysis using the KHB module of STATA [58]. This method can be used to decompose the overall effect into direct and indirect effects (i.e., mediational effect), and to estimate the mediating proportions of the total effect, that is, the ratio of the indirect effect to the total effect. In other words, this method compares the full model with a reduced model that replaces the mediators by the residuals of the mediators from a regression of the mediators on the key independent variable. This method was developed by Karlson, Holm, and Breen (KHB) [59]. We used this method because our outcome variable is binary, and this method is enough capable to extend the decomposition nature of linear models to nonlinear probability models. It allows decomposition of total effect into direct and indirect effects. The KHB method is a general decomposition method that is

unaffected by the rescaling or attenuation bias that arises in cross-model comparisons in nonlinear models. The KHB method works by residualizing the mediator, Z, by regressing it on X using OLS, and then another nonlinear model is estimated:

$$logit[pr(Y = 1)] = \frac{\beta_{YX.Z} + \beta_{YZ.X}\theta_{ZX}}{\sigma_e} X,$$

Where $\beta_{YX}$ is the total effect of x on y; $\beta_{YZ.X}$ is the partial effect of $Z$ on $Y$ given $X$; $\theta_{ZX}$ is a term from an auxiliary regression of Z on X and $\sigma_e$ is a scale parameter which allows the variance of the error to differ from that of the standard logistic distribution [60]. Below given the formation of decomposition based on "product coefficient method":

$$\text{Direct effect}: b_{YX.Z} = \frac{\beta_{YX.Z}}{\sigma_e},$$

$$\text{Indirect effect}: \theta_{ZX}b_{YZ.X} = \frac{\theta_{ZX} \times \beta_{YZ.X}}{\sigma_e},$$

$$\text{Total effect}: \frac{\beta_{YX}}{\sigma_e} = \frac{\beta_{YX.Z} + \theta_{ZX} \times \beta_{YZ.X}}{\sigma_e},$$

The total effect shows the impact of the exposure variable on the outcome variable without controlling any mediating variable. The direct effect represents the impact of the exposure variable on the outcome variable with the inclusion of mediators as controls. The indirect effect shows the impact of the exposure variable on the outcome variable through the mediators. In our study context, the indirect effect provides the extent to which multimorbidity is associated with selected mediators, and the extent to which the selected mediators is associated with the depression. We used mediating percentage only when the indirect effect is statistically significant [61]. Each potential mediating variable was tested in the model individually. All the models in mediation analysis were adjusted for the eight socio-demographic variables. Moreover, each model was tested independently for older men and women. National individual sample weight was used in the analysis. The level of statistically significance was set at two-sided p-value of 0.05. We performed all the analysis using Stata version 16.0 (Stata Corp LP, College Station, Texas).

## Results

### Socio-demographic profile of the study participants

Table 1 shows the socio-demographic profiles, functional health, and behaviour of individuals aged 60 years above, about 47 percent of them are men, and the remaining are women. A higher proportion (58.5%) of elders are of age 60–69 years. Nearly 59 percent of women are in the age group 60–69 years while their representation declines to 11.5 percent in age 80 years and above. Approximately 70 percent of the individuals were from a rural area (men– 72.1% and women– 69.2%). More than half of the older people had no educational status, the proportion of women (72.7%) in this category is higher than men (56.5%). Only 14.2 percent of the older people had ten and above years of education. A significantly higher percentage of men (82.3%) were in a union than women (44.3%). Women experienced more problems in sleep (40.3%), ADL (26.3%), and IADL (54.5%) than men. However, the prevalence of drinking alcohol (15.8%) and currently smoking (25.1%) was higher in men. About one-fourth of

**Table 1. Socio-demographic and health-related profile of the study sample by gender, LASI 2017–18.**

| Variables | Category | Total (n = 31,464) | Men (n = 15,098) | Women (n = 16,366) | P-Value |
|---|---|---|---|---|---|
| | | n (%) | n (%) | n (%) | |
| *Outcome Variable* | | | | | |
| **Depression** | No | 22958 (71.03) | 11377 (73.76) | 11581 (68.58) | <0.0001 |
| | Yes | 8506 (28.97) | 3721 (26.24) | 4785 (31.42) | |
| *Exposure Variable* | | | | | |
| **Multimorbidity** | No | 23853 (76.47) | 11630 (78) | 12223 (75.09) | <0.0001 |
| | Yes | 7521 (23.53) | 3406 (22) | 4115 (24.91) | |
| *Sociodemographic variables* | | | | | |
| **Age** | Aged 60–69 | 18974 (58.51) | 8961 (57.82) | 10013 (59.13) | <0.0001 |
| | Aged 70–79 | 9101 (30.2) | 4545 (31.14) | 4556 (29.13) | |
| | Aged over 80 | 3389 (11.29) | 1592 (11.04) | 1797 (11.52) | |
| **Residence** | Urban | 10739 (29.45) | 5021 (27.95) | 5718 (30.82) | 0.002 |
| | Rural | 20725 (70.55) | 10077 (72.05) | 10648 (69.18) | |
| **Education** | No Education | 16889 (56.52) | 5479 (38.6) | 11410 (72.7) | <0.0001 |
| | less than 5 years | 3781 (11.44) | 2184 (14.52) | 1597 (8.65) | |
| | 5–9 years | 6017 (17.83) | 3850 (24.11) | 2167 (12.16) | |
| | 10 or more years | 4777 (14.21) | 3585 (22.78) | 1192 (6.48) | |
| **Marital Status** | Currently in union | 20090 (62.09) | 12506 (81.72) | 7584 (44.36) | <0.0001 |
| | Not in union | 11374 (37.91) | 2592 (18.28) | 8782 (55.64) | |
| **Caste** | SC/ST | 10313 (27.76) | 4884 (27.17) | 5429 (28.29) | 0.150 |
| | OBC | 11886 (46.45) | 5781 (47.03) | 6105 (45.93) | |
| | Others | 8218 (25.79) | 3970 (25.8) | 4248 (25.78) | |
| **Religion** | Hindu | 23037 (82.31) | 11078 (82.12) | 11959 (82.48) | 0.496 |
| | Muslim | 3731 (11.29) | 1804 (11.73) | 1927 (10.89) | |
| | Others | 4630 (6.4) | 2186 (6.14) | 2444 (6.63) | |
| **MPCE Quintile** | Poor | 12961 (43.42) | 6103 (42.15) | 6858 (44.56) | 0.007 |
| | Middle | 6416 (20.95) | 3064 (21.6) | 3352 (20.35) | |
| | Rich | 12087 (35.64) | 5931 (36.24) | 6156 (35.09) | |
| *Functional health* | | | | | |
| **ADL disability** | No | 24770 (76.41) | 12377 (79.41) | 12393 (73.7) | <0.0001 |
| | Yes | 6694 (23.59) | 2721 (20.59) | 3973 (26.3) | |
| **IADL disability** | No | 18171 (53.83) | 10002 (63.11) | 8169 (45.46) | <0.0001 |
| | Yes | 13293 (46.17) | 5096 (36.89) | 8197 (54.54) | |
| **Poor sleep** | No | 20448 (63.67) | 10477 (68.13) | 9971 (59.69) | <0.0001 |
| | Yes | 10910 (36.33) | 4554 (31.87) | 6356 (40.31) | |
| **Pain** | No | 12686 (60.41) | 5317 (34.67) | 7369 (43.99) | <0.0001 |
| | Yes | 18669 (39.59) | 9712 (65.33) | 8957 (56.01) | |
| **SRH** | Good | 23785(75.79) | 11691 (77.75) | 11994 (74.03) | <0.0001 |
| | Poor | 7113(24.21) | 3087 (22.25) | 4026 (25.97) | |
| *Behavioural health* | | | | | |
| **Current smoker** | No | 27090 (86.34) | 11273 (74.95) | 15660 (96.63) | <0.0001 |
| | Yes | 4374 (13.66) | 3668 (25.05) | 706 (3.37) | |
| **Alcohol use** | No | 28324 (91.23) | 12447 (84.11) | 15847 (97.67) | <0.0001 |
| | Yes | 3140 (8.77) | 2621 (15.89) | 519 (2.33) | |
| **Physical inactivity** | No | 9545 (30.76) | 5853 (40.21) | 3692 (22.24) | <0.0001 |
| | Yes | 21899 (69.24) | 9234 (59.79) | 12665 (77.76) | |

(*Continued*)

**Table 1.** (Continued)

| Variables | Category | Total (n = 31,464) | Men (*n* = 15,098) | Women (*n* = 16,366) | P-Value |
|---|---|---|---|---|---|
| | | *n (%)* | *n (%)* | *n (%)* | |
| **Obesity** | No | 26417 (94.52) | 13065 (97.21) | 13352 (92.09) | <0.0001 |
| | Yes | 1633 (5.48) | 444 (2.79) | 1189 (7.91) | |

SC: Scheduled caste; ST: Scheduled tribe; OBC: Other backward class; MPCE: Monthly Per Capita Expenditure; SRH: Self-rated health; ADL: Activities of Daily Living; IADL: Instrumental activities of daily living; Frequency (n) are unweighted and percentage (%) are weighted; The sample may differ as all older adults did not give consent for measurement.

women had multimorbidity, while only 22% of men had multimorbidity. The prevalence of depression in older men was 26.2% while among older women this was around 31.4 percent.

## The association between multimorbidity and depression

Depression prevalence markedly increased with multimorbidity status; depression prevalence increased from 25.2% among older men without multimorbidity to 31.2% among older men with multimorbidity and from 30.4% among women without multimorbidity to 34.5 percent among women with multimorbidity (Fig 2). Overall, 27.9% of the older people without multi-morbidity condition were experiencing depression, while 33.1% of the older people with multimorbidity were depressed.

Table 2 shows the results of multivariable binary logistic regression models. The table also reports the separate models for the older women and men. In unadjusted analysis (model 1), we found that older adults with multimorbidity have a significantly higher probability of depression.

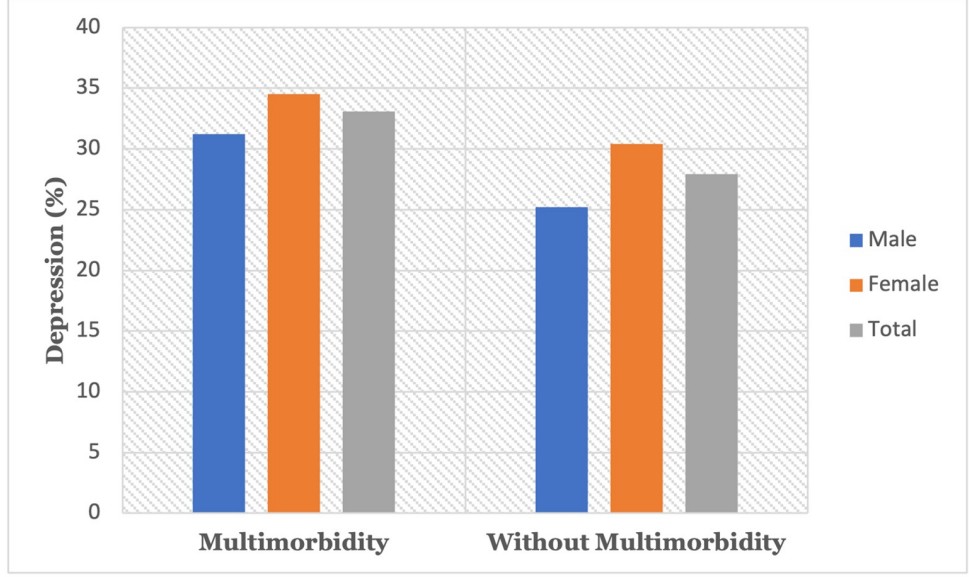

**Fig 2. Prevalence of depression across the multimorbidity status in older adults.**

**Table 2. Associations between multimorbidity and depression among older adults: Logistic regression models.**

| Outcome: Depression | Model 1 | | | Model 2 | | | Model 3 | | |
|---|---|---|---|---|---|---|---|---|---|
| | UOR | 95% CI | Pseudo R2 | AOR | 95% CI | Pseudo R2 | AOR | 95% CI | Pseudo R2 |
| **Main model** | | | | | | | | | |
| Multimorbidity | 1.28*** | 1.13–1.44 | 0.002 | 1.43*** | 1.27–1.61 | 0.018 | 1.27*** | 1.12–1.45 | 0.043 |
| **Stratified models** | | | | | | | | | |
| Men: Multimorbidity | 1.35*** | 1.15–1.58 | 0.003 | 1.54*** | 1.30–1.82 | 0.020 | 1.40*** | 1.15–1.69 | 0.043 |
| Women: Multimorbidity | 1.21*** | 1.01–1.44 | 0.001 | 1.36*** | 1.16–1.59 | 0.015 | 1.19* | 1.00–1.41 | 0.045 |

UOR: Unadjusted Odds Ratio; AOR: Adjusted Odds Ratio; CI: Confidence interval; Pseudo R$^2$: Measure of model fitting on the same data, predicting the same outcome.

Model 1: Unadjusted model

Model 2: Adjusted for age, gender, marital status, residence, education level, religion, caste, MPCE quintile.

Model 3: Adjusted for Model 2, ADL disability, IADL disability, poor sleep, pain, SRH, currently smoking, alcohol use, Physical Inactivity, and obesity.

Gender-stratified models were not adjusted for gender.

*p < 0.05;

**p < 0.005;

***p < 0.001

While the odd ratio is significantly higher for older men (UOR:1.35, 95% CI: 1.15–1.58) than older women (UOR:1.21, 95% CI: 1.01–1.44). After adjusting socio-demographic and potential mediators, we found that multimorbidity is still significantly associated with a higher probability of depression among older adults, with odds ratios of 1.27 (95% CI: 1.12–1.45) and odds of being depressed is significantly higher for older men than women.

## Explaining the association between multimorbidity and depression

Table 3 represent the mediation statistics for the overall sample. Multimorbidity-depression association was mediated by functional health such as SRH, poor sleep, ADL and IADL disability, and pain. The mediating percentage ranged between 8% and 40% respectively for all described mediators.

Only the physical inactivity among all behavioural health variables mediated the association between multimorbidity and depression and the mediating proportion is only 5.19%. Table 4 shows the results of the mediation analysis for older men.

For older men, mediation analysis shows that multimorbidity has an indirect effect on depression through factors related to functional health including ADL disability, IADL disability, poor sleep, SRH, and pain. Mediating percentage in the association was high for SRH (38.05%) followed by poor sleep (29.47%), IADL disability (21.98%), and ADL disability (21.21%). Pain, on the other hand, can be seen as a mediator, but with a lower mediating percentage of 5.79% among functional health variables. Moreover, among factors related to behavioural health, physical inactivity (5.11%) was the only significant mediator of the association. In contrast, smoking, alcohol, and obesity are not significant mediators for the association between multimorbidity and depression among older men.

Table 5 presents the results of mediation statistics for older women. For older women, multimorbidity has an indirect effect on depression through ADL disability (21.90%), IADL disability (23.32%), pain (9.96%), poor sleep (40.52%), and SRH (40.17%). Meanwhile, physical inactivity mediated 5.17% of the association between multimorbidity and depression. Whereas smoking, alcohol, and obesity are not significantly mediators for the association among older women.

**Table 3. Functional and behavioural mediators in the association between multimorbidity and depression (based on total sample).**

| Mediators | Indirect effect | | | Direct effect | | | Total effect | | | Mediating % |
|---|---|---|---|---|---|---|---|---|---|---|
| | OR | 95% CI | P-Value | OR | 95% CI | P-Value | OR | 95% CI | P-Value | |
| *Functional health* | | | | | | | | | | |
| ADL disability | 1.07 | [1.06,1.09] | <0.001 | 1.30 | [1.22,1.38] | <0.001 | 1.40 | [1.31,1.48] | <0.001 | 21.49 |
| IADL disability | 1.08 | [1.07,1.09] | <0.001 | 1.30 | [1.22,1.38] | <0.001 | 1.40 | [1.32,1.49] | <0.001 | 22.65 |
| Poor sleep | 1.13 | [1.11,1.14] | <0.001 | 1.25 | [1.17,1.33] | <0.001 | 1.40 | [1.32,1.49] | <0.001 | 35.15 |
| Pain | 1.03 | [1.02,1.03] | <0.001 | 1.36 | [1.28,1.45] | <0.001 | 1.40 | [1.32,1.49] | <0.001 | 7.92 |
| SRH | 1.15 | [1.13,1.16] | <0.001 | 1.23 | [1.16,1.31] | <0.001 | 1.42 | [1.33,1.51] | <0.001 | 40.0 |
| *Behavioural health* | | | | | | | | | | |
| Currently smoking | 1.00 | [1,1] | 0.047 | 1.40 | [1.32,1.49] | <0.001 | 1.40 | [1.31,1.48] | <0.001 | NA |
| Alcohol use | 1.00 | [1,1] | 0.052 | 1.39 | [1.31,1.48] | <0.001 | 1.40 | [1.31,1.48] | <0.001 | NA |
| Physical inactivity | 1.02 | [1.01,1.02] | <0.001 | 1.39 | [1.31,1.48] | <0.001 | 1.42 | [1.34,1.51] | <0.001 | 5.19 |
| Obesity | 0.99 | [0.99,1] | 0.124 | 1.42 | [1.33,1.51] | <0.001 | 1.41 | [1.32,1.5] | <0.001 | NA |

OR: Odds ratio; CI: Confidence Interval; SRH: Self-rated health; ADL: Activities of Daily Living; IADL: Instrumental activities of daily living. Models are adjusted for age, gender, education, place of residence, marital status, caste, religion, and MPCE quintile.

## Discussion

The current study was based on the nationally representative data of LASI, Wave-1, to examine the sex differences in the association between multiple co-occurring illnesses and depression, including the mediating role of functional and behavioural health. The study showed that multimorbidity condition was associated with depression in both men and women among older adults, even after adjustment of socio-demographic and mediating variables included in the study. While results from mediation analysis revealed that functional health mediates the association between multimorbidity and depression across the gender.

Noticeably, the association between multimorbidity and depression has been explored over the last two decades. However, there are conflicting findings of the direct link between multimorbidity and depression until and unless one individual experiences the discomfort

**Table 4. Functional and behavioural mediators in the association between multimorbidity and depression (Men).**

| Mediators | Indirect effect | | | Direct effect | | | Total effect | | | Mediating % |
|---|---|---|---|---|---|---|---|---|---|---|
| | OR | 95% CI | P-Value | OR | 95% CI | P-Value | OR | 95% CI | P-Value | |
| *Functional health* | | | | | | | | | | |
| ADL disability | 1.08 | [1.06,1.1] | <0.001 | 1.33 | [1.22,1.46] | <0.001 | 1.44 | [1.31,1.58] | <0.001 | 21.21 |
| IADL disability | 1.08 | [1.07,1.1] | <0.001 | 1.33 | [1.21,1.46] | <0.001 | 1.44 | [1.32,1.58] | <0.001 | 21.98 |
| Poor sleep | 1.12 | [1.09,1.14] | <0.001 | 1.30 | [1.18,1.42] | <0.001 | 1.45 | [1.32,1.59] | <0.001 | 29.47 |
| Pain | 1.02 | [1.01,1.03] | <0.001 | 1.41 | [1.29,1.55] | <0.001 | 1.45 | [1.32,1.58] | <0.001 | 5.79 |
| SRH | 1.16 | [1.13,1.18] | <0.001 | 1.27 | [1.15,1.39] | <0.001 | 1.46 | [1.33,1.6] | <0.001 | 38.05 |
| *Behavioural health* | | | | | | | | | | |
| Currently smoking | 1.00 | [0.99,1] | 0.07 | 1.45 | [1.32,1.59] | <0.001 | 1.44 | [1.32,1.58] | <0.001 | NA |
| Alcohol use | 1.00 | [1,1] | 0.11 | 1.44 | [1.31,1.58] | <0.001 | 1.44 | [1.32,1.58] | <0.001 | NA |
| Physical inactivity | 1.02 | [1.01,1.03] | <0.001 | 1.41 | [1.30,1.55] | <0.001 | 1.44 | [1.32,1.58] | <0.001 | 5.11 |
| Obesity | 0.99 | [0.98,1] | 0.05 | 1.46 | [1.32,1.61] | <0.001 | 1.45 | [1.31,1.59] | <0.001 | NA |

OR: Odds ratio; CI: Confidence Interval; SRH: Self-rated health; ADL: Activities of Daily Living; IADL: Instrumental activities of daily living. Models are adjusted for age, education, place of residence, marital status, caste, religion, and MPCE quintile.

**Table 5. Functional and behavioural mediators in the association between multimorbidity and depression (Women).**

| Mediators | Indirect effect | | | Direct effect | | | Total effect | | | Mediating % |
|---|---|---|---|---|---|---|---|---|---|---|
| | OR | 95% CI | P-Value | OR | 95% CI | P-Value | OR | 95% CI | P-Value | |
| *Functional health* | | | | | | | | | | |
| ADL disability | 1.07 | [1.06,1.08] | <0.001 | 1.27 | [1.17,1.38] | <0.001 | 1.36 | [1.25,1.48] | <0.001 | 21.90 |
| IADL disability | 1.08 | [1.06,1.09] | <0.001 | 1.27 | [1.17,1.38] | <0.001 | 1.37 | [1.26,1.48] | <0.001 | 23.32 |
| Poor sleep | 1.14 | [1.11,1.16] | <0.001 | 1.21 | [1.11,1.31] | <0.001 | 1.37 | [1.26,1.49] | <0.001 | 40.52 |
| Pain | 1.03 | [1.02,1.04] | <0.001 | 1.32 | [1.22,1.43] | <0.001 | 1.36 | [1.26,1.48] | <0.001 | 9.96 |
| SRH | 1.14 | [1.12,1.16] | <0.001 | 1.21 | [1.11,1.32] | <0.001 | 1.38 | [1.27,1.5] | <0.001 | 40.17 |
| *Behavioural health* | | | | | | | | | | |
| Currently smoking | 1.00 | [1,1] | 0.90 | 1.36 | [1.25,1.48] | <0.001 | 1.36 | [1.25,1.48] | <0.001 | NA |
| Alcohol use | 1.00 | [1,1] | 0.18 | 1.36 | [1.25,1.47] | <0.001 | 1.36 | [1.25,1.48] | <0.001 | NA |
| Physical inactivity | 1.02 | [1.01,1.02] | <0.001 | 1.38 | [1.27,1.49] | <0.001 | 1.40 | [1.30,1.52] | <0.001 | 5.17 |
| Obesity | 1.00 | [0.98,1.01] | 0.59 | 1.38 | [1.27,1.5] | <0.001 | 1.38 | [1.26,1.5] | <0.001 | NA |

OR: Odds ratio; CI: Confidence Interval; SRH: Self-rated health; ADL: Activities of Daily Living; IADL: Instrumental activities of daily living. Models are adjusted for age, education, place of residence, marital status, caste, religion, and MPCE quintile.

concurrently due to health-related functional and emotional issues [62]. More importantly, our hypothesis regarding the possible mediating role of factors related to functional health in the association of multimorbidity with depression was supported, but it was not favorable in the case of behavioural health.

Results from mediation analysis revealed that functional health issues play a crucial role in the indirect association between multimorbidity and depression in both men and women. Our study findings also align with existing research showing that functional limitations, poor sleep, and chronic pain are all prominent symptoms as individuals get older, especially in those who have several chronic conditions, possibly crucial for the pathway of multimorbidity-depression association [26, 63–66].

Furthermore, SRH was found to be the strongest mediator of the examined factor in the study and has an indirect effect on the incidence of depression among older men and women who experience multimorbidity. In line with previous research, we found that diagnosing one or more chronic illnesses increases the risk of reporting poorer self-rated health at a later age [67, 68], and depression was strongly associated with poor self-rated health [69]. This might explain why self-reported health is an important factor in the association between multimorbidity and depression. In older adults, SRH has shown to be an important indicator of disease effect on an individual's wellbeing [70]. As documented, SRH as a construct of overall subjective health should be given more attention in diagnosing health problems among older which is also evident from the present study [53, 71].

Poor sleep was found to be second strongest mediator among the variables used in the study. This may indicate that sleep quality or sleep behaviour play a unique role in the multimorbidity-depression association. One of the possible explanations for this finding is that sleep disruption is one of the most prevalent symptoms of depression in older people [72, 73], and it occurs as a result of the emergence of several chronic diseases [74]. Poor sleep, multimorbidity, and depressive symptoms all might have a bidirectional or reciprocal link that influences one another, however, further longitudinal study are required to make any causal inference.

We also found that ADL and IADL mediated the association between multimorbidity and depression. Existing research has already shown that after diagnosing a high number of

chronic illnesses, an individual may experience functional limitation and disability, leading to poor quality of life [75]. Functional and social limitations that lead to constant assistance for daily tasks, mobility, and accessing the healthcare facilities in the later life of individuals. Thus, the multimorbidity conjoining with these consequences might further strongly associated with the depression [24, 76]. Therefore, the present study finding suggests that functional limitations should be considered when exploring the underlying mechanisms that connect multimorbidity with depression.

Surprisingly, factors related to behavioural health did not appear to mediate the link between multimorbidity and depressive symptoms, except physical inactivity. Therefore, smoking status, alcohol consumption, and obesity have no significant indirect effect on the multimorbidity-depression association among older adults. The addition hypothesis provides a plausible explanation for behavioural health's insignificant mediation effect in the connection between multimorbidity and depressed symptoms. Concerning health-risk activities, most notably smoking and alcohol use, addiction theories claimed that smoking and alcohol intake may provide temporary mental satisfaction, tempting even senior men and women into such intoxication [77–80]. At a same time, a study found that health comprising behavior such as alcohol, daily smoking, and unhealthy diet has little importance in the association between loneliness and multiple chronic illnesses in older people [81].

This study may support our findings because loneliness is one of the symptoms of depression. To our knowledge, no prior study explored behavioural health as a mediator of the specific association between multimorbidity and depression in old age. However, more research is needed to temper the claim of certainty. However, physical inactivity was found to mediate the association between multimorbidity and depression. These findings confirm previous findings, which show that multimorbidity is an independent risk factor for physical inactivity [82] and is associated with depression. Therefore, physical inactivity associated with multimorbidity might play an important role in depression in older adults.

Regarding gender differences in the association, the odds ratios for depression are slightly higher among older men with multimorbidity than older women, suggesting that older men were more sensitive to chronic diseases and hence more likely to have depressive symptoms. In older individuals, there are disparities between men and women in behavioural and biological health, with men having more chronic fatal or life-threatening illnesses, which might contribute to increased depressive symptoms [83]. However, all potential mediators were found to be stronger for older women than for men. Sex disparities in chronic morbidity and, as a result of which women have less ability to perform everyday activities and have poor cognitive and mental health, might be one explanation for the current findings [84, 85]. This finding suggests a public health intervention to improve the mental health of older adults who experiences the number of chronic illnesses/multimorbidity that would be responsive to sex differences.

## Limitations

The strengths of this study include a large nationally representative sample of older adults the evaluation of functional and behavioural health, which allowed for a detailed description of the research population, adjustments to the analyses, and the investigation of several possible mediators. Our study also met with several limitations. Firstly, the cross-sectional design of this study does not necessarily infer a causal relationship between multimorbidity and depression. Secondly, information on morbidity and lifestyle behaviour was self-reported, leading to biases in estimating the prevalence of morbidities.

## Conclusions

The findings of this population-based study revealed that multimorbid older people are more likely to suffer depressive symptoms, suggesting the need for more chronic disease management and research. The current finding implies that clinicians should pay attention to different functional health and health behaviour when treating multimorbidity, depression in older people. Further, the study revealed a gender difference in the association between geriatric depression and multimorbidity in India. Although chronic illness cannot usually be cured entirely, although it may be managed by altering one's lifestyle and treatment process. Thus, functional health especially among women should be a priority for primary care physicians during severe chronic disease diagnosis processes. However, because there might be a bi-directional association between multimorbidity and depression, further longitudinal research is needed to understand the mechanism of functional and behavioural health in multimorbidity-depression relationship. The recommendations for preventing and treating chronic illnesses and depression in older adults should be strengthened in light of gender differences.

## Supporting information

**S1 Appendix.**
(DOCX)

## Author Contributions

**Conceptualization:** Salmaan Ansari, Abhishek Anand.

**Data curation:** Abhishek Anand.

**Formal analysis:** Salmaan Ansari.

**Methodology:** Salmaan Ansari.

**Writing – original draft:** Salmaan Ansari, Abhishek Anand, Babul Hossain.

**Writing – review & editing:** Babul Hossain.

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
