## [Decision Letter · Decision Letter 0]

16 Mar 2022

PONE-D-22-00909Multimorbidity and depression among older adults in India: Mediating role of functional and behavioural healthPLOS ONE

Dear Dr. Anand,

Thank you for submitting your manuscript to PLOS ONE. After careful consideration, we feel that it has merit but does not fully meet PLOS ONE’s publication criteria as it currently stands. Therefore, we invite you to submit a revised version of the manuscript that addresses the points raised during the review process.

Thank you for your patience while waiting for reviews of your manuscript. As you can see, we have now received 2 reviews of your work - both of which are quite detailed in their suggestions to improve the manuscript. You will note that the reviewers made a number of suggestions about strengthening the rationale for this research and the final discussion; as well, they made a number of suggestions regarding the analysis that I think it would be important to address.

We look forward to receiving your revised manuscript.

Kind regards,

Andrea Gruneir

Academic Editor

PLOS ONE

Journal Requirements:

Reviewers' comments:

Reviewer's Responses to Questions

**Comments to the Author**

1. Is the manuscript technically sound, and do the data support the conclusions?

Reviewer #1: Yes

Reviewer #2: Partly

2. Has the statistical analysis been performed appropriately and rigorously? 

Reviewer #1: Yes

Reviewer #2: No

3. Have the authors made all data underlying the findings in their manuscript fully available?

Reviewer #1: Yes

Reviewer #2: No

4. Is the manuscript presented in an intelligible fashion and written in standard English?

Reviewer #1: Yes

Reviewer #2: No

5. Review Comments to the Author

Reviewer #1: The research topic is interesting. Use of LASI is the major strength of the study. However, there are few concerns:

1. Were survey weights used in the analysis?

2. The study highlights both direct and indirect pathways between multimorbidity and depression. However, merging healthy individual with individuals with single morbidity is not reassuring, it might have reflected poorly on the study estimates. If possible, try using mm as a categorical variable (with three distinct category).

3. The authors are requested to use standard terminology. Use of word femen is not a standard generally used in research articles.

4. Try to add more background on the point of gender differences, as it is highlighted as the major research objective in the discussion section.

5. Authors have not properly reviewed the literature, Sanghamitra Pati and GK Mini are the two authors which needs to be cited in any article published on multimorbidity in India. Thus might reflect poorly to the international readers. Try citing articles written by researchers outside IIPS to add more weightage to your study.

6. Abstract uses the term univariate logistic regression model. What does that supposed to mean. For regression analysis it should be binary. If this statement means anything else, than consider reframing the sentence to sound more clear.

The article can be accepted from my end.

Reviewer #2: The study “Multimorbidity and depression among older adults in India: Mediating role of functional and behavioural health” is generally well developed. It examined the influence of multimorbidity on elders’ depression and its gender disparities while assessing functional health’s and behavioral health’s mediating effects. Finally, the study revealed that multimorbid older people are more likely to suffer depressive symptoms, and the association may be mediated by certain functional health factors, especially in older women. Below are some comments for the authors’ consideration.

In the Introduction section:

1. Some hypothesis that should be required to further verified and proved by the study were stated in affirmative tone, which is obviously inappropriate. For instance, lines 90 to 91 “It is also one of the major predictors of mental health of older population.”

2. It has been well told functional health could be a mediating factor between multimorbidity and depression, so it is necessary to inform that why behavioral health could also be a mediating factor. The statement in lines 99 to 101, page 4 is not sufficient.

3. The covariates, selected in Figure 1, should be supported fully by theoretical or empirical evidence. For example, some specific indicators of covariates and functional health, such as education, income, ADL, IADL, etc., are not mentioned in the introduction section. So, it is recommended to supplement more evidences on those items to build the theoretical model.

4. Besides, to highlight that “Age-related multimorbidity is an escalating issue that poses a major challenge for health care systems worldwide” fully, it is better to add some evidences from developed countries, but not only from low-middle income countries.

In the Methods section:

5. The activities of ADL and IADL are counted six and seven respectively, in lines 199 -205, on page 8, while it is said “a series of five and six different questions” in lines 198 -199. It is not consistent.

6. It is not clear for the code of some variables. For example, Is never and rarely sleep coded as 0, occasionally and frequently sleep coded as 1? Pain and Self-rated health are also the same. It is recommended to introduce variables’ coding way clearly.

In the Results section:

7. The statistical difference among the subgroups of various factors should be reported in Table 1.

8. It is better to add statement for the meaning of “UOR, AOR, 95%CI, and Pseudo R2” clearly below the table 2.

9. It is suggested to add the results about covariates in each model in table 2, 3, 4, and 5, which will be helpful for showing the impact of demographic characteristics on elders’ depression.

10. It is recommended to list the indirect effect value, direct effect value and total effect value in table 3, 4 and 5, and add a note to explain the calculation formula of these effect values below the table 3.

11. It will be better to elaborate the results in tables 4 and 5 separately, and the lines 365-366, on page 16.

12. There were some confused issues. Why the mediating % in table 4 and table 5 do not match the data in the text (lines 384-385, lines 387-389, on page 17)? Why is there still a mediating % of physical inactivity shown in table 5, but the text stated that physical inactivity isn’t a mediator between multimorbidity and depression for older women. (lines 389-390, on page 18).

13. Does the different mediating % reflect distinct impact degrees of different factors on the association between multimorbidity and depression for elders? However, the paper only stated the mediating % without further comparison, it feels leaving out more valuable information.

In the Discussion section:

14. From the tables in results section, the mediating % of poor sleep in the overall sample and male sample are both the second, so, how to judge poor sleep was the strongest mediator of the examined factor in the study (lines 412- 413, on page 13)? Which is the same in lines 422 to 424, on page 13. The mediating % from the total sample and male sample show that self-rated health are both the largest, so, why is self-rated health the second stronger mediator among the variables used in the study?

15. The sentence “Poor sleep, multimorbidity,…that influences one another.” in lines 418-419, on page 13 is too affirmative, for the study did not explore the bidirectional effect between factors.

16. What does the“dis”in line 431, on page 19 refer to?

17. Grammar error in line 432, on page 19 “leading leads to poor quality of life”.

18. “the need for healthcare facilities…depression in older adults.” in lines 433- 435, on page 19, this explanation is too general and abstract to make it difficult to understand.

19. The discussion section mentions the results repeatedly, but doesn’t explain the deep reasons behind the results. So, the reasons of the results are not fully discussed.

In the Conclusions section:

20. The recommendations for prevention and treatment of chronic diseases and depression among elders should be further supplemented from perspective of gender disparities.

6. PLOS authors have the option to publish the peer review history of their article (what does this mean?). If published, this will include your full peer review and any attached files.

Reviewer #1: **Yes: **Parul Puri

Reviewer #2: No

---

## [Author Response · Author response to Decision Letter 0]

16 Apr 2022

Reviewer #1: The research topic is interesting. Use of LASI is the major strength of the study. However, there are few concerns:

1. Were survey weights used in the analysis?

Response – Yes, We have mentioned this in methodology section.

2. The study highlights both direct and indirect pathways between multimorbidity and depression. However, merging healthy individual with individuals with single morbidity is not reassuring, it might have reflected poorly on the study estimates. If possible, try using mm as a categorical variable (with three distinct category).

Response –Dear reviewer, thanks for the kind words and acknowledging the importance of the theme of our manuscript. Even though, previous literature has shown that living with multiple health conditions is even more impactful than any one health condition, particularly in later life. Due to treatment burden and poor disease management, mental health disorders, particularly depression, are more prevalent in persons with an increasing number of physical disorders. We agree with the reviewer, but taking mm as a category variable was beyond the scope of our analysis in broader approach, but would be useful in future research.

3. The authors are requested to use standard terminology. Use of word femen is not a standard generally used in research articles.

Response – Thank you for noticing. Comments are incorporated throughout the manuscript.

4. Try to add more background on the point of gender differences, as it is highlighted as the major research objective in the discussion section.

Response – Thank you for your suggestion, we have added few more points on gender differences.

5. Authors have not properly reviewed the literature, Sanghamitra Pati and GK Mini are the two authors which needs to be cited in any article published on multimorbidity in India. Thus might reflect poorly to the international readers. Try citing articles written by researchers outside IIPS to add more weightage to your study.

Response – Thank you for suggestions, we have incorporated your suggestions.

6. Abstract uses the term univariate logistic regression model. What does that supposed to mean. For regression analysis it should be binary. If this statement means anything else, than consider reframing the sentence to sound more clear.

Response – Dear reviewer, thanks for guiding. Yes, the statement is modified.

The article can be accepted from my end.

Reviewer #2: The study “Multimorbidity and depression among older adults in India: Mediating role of functional and behavioural health” is generally well developed. It examined the influence of multimorbidity on elders’ depression and its gender disparities while assessing functional health’s and behavioral health’s mediating effects. Finally, the study revealed that multimorbid older people are more likely to suffer depressive symptoms, and the association may be mediated by certain functional health factors, especially in older women. Below are some comments for the authors’ consideration.

In the Introduction section:

1. Some hypothesis that should be required to further verified and proved by the study were stated in affirmative tone, which is obviously inappropriate. For instance, lines 90 to 91 “It is also one of the major predictors of mental health of older population.”

Response – Thank you for suggestion. We modified the statement accordingly.

2. It has been well told functional health could be a mediating factor between multimorbidity and depression, so it is necessary to inform that why behavioral health could also be a mediating factor. The statement in lines 99 to 101, page 4 is not sufficient.

Response – Thank you for suggestion, we have added more information on behavioural health part.

3. The covariates, selected in Figure 1, should be supported fully by theoretical or empirical evidence. For example, some specific indicators of covariates and functional health, such as education, income, ADL, IADL, etc., are not mentioned in the introduction section. So, it is recommended to supplement more evidences on those items to build the theoretical model.

Response – Thank you for your suggestion. We incorporated the suggestion for ADL and IADL in the revised manuscript. As this study mainly focused on the multimorbidity and depression status considering the mediating role functional health among older adults, we have tried to cover the existing literature focusing on these aspects. While education, income and other covariates are controlled in our study which are not main focus of the study. These covariates were included in the model to ensure that these variables did not confound any of the assessed direct and indirect associations between multimorbidity status and depression. Thus, we have intentionally skipped the discussion on covariates. A separate study can be caried out to highlight mentioned factors role in multimorbidity and depression. 

4. Besides, to highlight that “Age-related multimorbidity is an escalating issue that poses a major challenge for health care systems worldwide” fully, it is better to add some evidences from developed countries, but not only from low-middle income countries.

Response – Evidences from developed countries added in the manuscript.

In the Methods section:

5. The activities of ADL and IADL are counted six and seven respectively, in lines 199 -205, on page 8, while it is said “a series of five and six different questions” in lines 198 -199. It is not consistent.

Response – Thank you for noticing. Yes, it is modified now.

6. It is not clear for the code of some variables. For example, Is never and rarely sleep coded as 0, occasionally and frequently sleep coded as 1? Pain and Self-rated health are also the same. It is recommended to introduce variables’ coding way clearly.

Response – Dear reviewer, thank you for suggestions. Now, coding for concerning variables has been introduced clearly in the revised manuscript. 

In the Results section:

7. The statistical difference among the subgroups of various factors should be reported in Table 1.

Response – The main objective of the Table 1 was to give the sample characteristics for the study population. Thus, the table only showed the percentage distribution of the total sample included in the study. As you suggested, we modified table 1 showing the statistical difference by gender for various factors. 

8. It is better to add statement for the meaning of “UOR, AOR, 95%CI, and Pseudo R2” clearly below the table 2.

Response – Thanks for suggestions. Comments are incorporated.

9. It is suggested to add the results about covariates in each model in table 2, 3, 4, and 5, which will be helpful for showing the impact of demographic characteristics on elders’ depression.

Response – Thank you for your suggestion. Table 2 highlighted the associations between multimorbidity and depression stratified for gender. Although there are several literatures focusing range of covariates of depression, we only aimed to examine the association multimorbidity and depression controlling covariates in Table 2, further adding the gender dimension. As you suggested and for the readers clarity, we have added these tables separately in the supplementary section. 

10. It is recommended to list the indirect effect value, direct effect value and total effect value in table 3, 4 and 5, and add a note to explain the calculation formula of these effect values below the table 3.

Response – Thank you for your suggestion. We had already given total, direct and indirect effect value in form of odds ratio. Further, the we have added a note to explain the calculation formula of these effect values in method section. 

11. It will be better to elaborate the results in tables 4 and 5 separately, and the lines 365-366, on page 16.

Response – Comments are incorporated. 

12. There were some confused issues. Why the mediating % in table 4 and table 5 do not match the data in the text (lines 384-385, lines 387-389, on page 17)? Why is there still a mediating % of physical inactivity shown in table 5, but the text stated that physical inactivity isn’t a mediator between multimorbidity and depression for older women. (lines 389-390, on page 18).

Response – Thank you for your observations. Now, the value of mediating percentage has been incorporated in the text according to the tables. 

13. Does the different mediating % reflect distinct impact degrees of different factors on the association between multimorbidity and depression for elders? However, the paper only stated the mediating % without further comparison, it feels leaving out more valuable information.

Response – Comments are incorporated. Comparisons of mediating percentage among functional and behavioral health has been added in the result section as well as discussion part.

In the Discussion section:

14. From the tables in results section, the mediating % of poor sleep in the overall sample and male sample are both the second, so, how to judge poor sleep was the strongest mediator of the examined factor in the study (lines 412- 413, on page 13)? Which is the same in lines 422 to 424, on page 13. The mediating % from the total sample and male sample show that self-rated health are both the largest, so, why is self-rated health the second stronger mediator among the variables used in the study?

Response – Thank you for noticing. Yes, self-rated health is the strongest mediators in the possible multimorbidity-depression association. We modified both the statements in discussion part accordingly. 

15. The sentence “Poor sleep, multimorbidity,…that influences one another.” in lines 418-419, on page 13 is too affirmative, for the study did not explore the bidirectional effect between factors.

Response – Thank you for your observations. Poor sleep, multimorbidity, and depressive symptoms all might have a bidirectional or reciprocal link that influences one another. However, further longitudinal study is required to make any causal inference. We also added this in the write up. 

16. What does the“dis”in line 431, on page 19 refer to?

Response – The typo has been corrected.

17. Grammar error in line 432, on page 19 “leading leads to poor quality of life”.

Response – The error has been resolved.

18. “the need for healthcare facilities…depression in older adults.” in lines 433- 435, on page 19, this explanation is too general and abstract to make it difficult to understand.

Response – As you suggested. Corrections have been done in this section. 

19. The discussion section mentions the results repeatedly, but doesn’t explain the deep reasons behind the results. So, the reasons of the results are not fully discussed.

Response – We have addressed the concerns in discussion section. 

In the Conclusions section:

20. The recommendations for prevention and treatment of chronic diseases and depression among elders should be further supplemented from perspective of gender disparities.

Response – Thank you for your observation. As suggested, we further highlighted the gender disparities in the conclusion part.

---

## [Decision Letter · Decision Letter 1]

11 May 2022

PONE-D-22-00909R1Multimorbidity and depression among older adults in India: Mediating role of functional and behavioural healthPLOS ONE

Dear Dr. Anand,

Thank you for submitting your manuscript to PLOS ONE. After careful consideration, we feel that it has merit but does not fully meet PLOS ONE’s publication criteria as it currently stands. Therefore, we invite you to submit a revised version of the manuscript that addresses the points raised during the review process.

The reviewers felt that your revisions were generally sufficient; however Reviewer 2 raised a few minor points that must be addressed. Their comments can be viewed below.

We look forward to receiving your revised manuscript.

Kind regards,

Natasha McDonald, PhD

Associate Editor

PLOS ONE

Journal Requirements:

Reviewers' comments:

Reviewer's Responses to Questions

**Comments to the Author**

1. If the authors have adequately addressed your comments raised in a previous round of review and you feel that this manuscript is now acceptable for publication, you may indicate that here to bypass the “Comments to the Author” section, enter your conflict of interest statement in the “Confidential to Editor” section, and submit your "Accept" recommendation.

Reviewer #1: All comments have been addressed

Reviewer #2: All comments have been addressed

2. Is the manuscript technically sound, and do the data support the conclusions?

Reviewer #1: Yes

Reviewer #2: Yes

3. Has the statistical analysis been performed appropriately and rigorously? 

Reviewer #1: Yes

Reviewer #2: Yes

4. Have the authors made all data underlying the findings in their manuscript fully available?

Reviewer #1: Yes

Reviewer #2: Yes

5. Is the manuscript presented in an intelligible fashion and written in standard English?

Reviewer #1: Yes

Reviewer #2: Yes

6. Review Comments to the Author

Reviewer #1: Thank you for the opportunity to review this interesting piece of work. The authors have done a great job in addressing the comments. There are only minor issues remaining in the paper, this includes grammatical, English language errors and references sections. In my opinion, this can be addressed in the proof-reading stage. I would recommend this article to be published after a rigorous proof-reading.

Reviewer #2: Overall, I think it is a good job for revising the manuscript and responding to previous comments. While I still have a few minor comments about this manuscript:

1. In the abstract, it seems to ignore mentioning the mediating proportion of the factor “pain”.

2. Why the mediating % of physical inactivity in table 3 does not match the data in the text (lines 402-404)?

3. In lines 471-473 of discussion section, the “a third factor” is confusing.

7. PLOS authors have the option to publish the peer review history of their article (what does this mean?). If published, this will include your full peer review and any attached files.

Reviewer #1: **Yes: **Parul Puri

Reviewer #2: No

---

## [Author Response · Author response to Decision Letter 1]

12 May 2022

Dear Editor

Thank you for your consideration. The reviewers have raised valid comments and we have incorporated those comments which have improved our manuscript. We have also prepared a point by point rebuttal to the comments raised by the reviewers.

Much regards

Comments to the Author

Reviewer #1: Thank you for the opportunity to review this interesting piece of work. The authors have done a great job in addressing the comments. There are only minor issues remaining in the paper, this includes grammatical, English language errors and references sections. In my opinion, this can be addressed in the proof-reading stage. I would recommend this article to be published after a rigorous proof-reading.

Reviewer #2: Overall, I think it is a good job for revising the manuscript and responding to previous comments. While I still have a few minor comments about this manuscript:

1. In the abstract, it seems to ignore mentioning the mediating proportion of the factor “pain”.

Response – Thank you for your comment, we have updated the abstract accordingly.

2. Why the mediating % of physical inactivity in table 3 does not match the data in the text (lines 402-404)?

Response – We have corrected this error in the manuscript.

3. In lines 471-473 of discussion section, the “a third factor” is confusing.

Response – Thank you for pointing out, we have corrected the sentence.

---

## [Editor Report · Decision Letter 2]

25 May 2022

Multimorbidity and depression among older adults in India: Mediating role of functional and behavioural health

PONE-D-22-00909R2

Dear Dr. Anand,

We’re pleased to inform you that your manuscript has been judged scientifically suitable for publication and will be formally accepted for publication once it meets all outstanding technical requirements.

Kind regards,

Carla Pegoraro

Division Editor

PLOS ONE

---

## [Editor Report · Acceptance letter]

30 May 2022

PONE-D-22-00909R2 

Multimorbidity and depression among older adults in India: Mediating role of functional and behavioural health 

Dear Dr. Anand:

I'm pleased to inform you that your manuscript has been deemed suitable for publication in PLOS ONE. Congratulations! Your manuscript is now with our production department. 

Kind regards, 

on behalf of

Dr Carla Pegoraro 

Staff Editor

PLOS ONE